# Pilot Implementation of Falsified Medicines Directive in Hospital Pharmacy to Develop Best Practices for Medicine Decommissioning Process

**DOI:** 10.3390/pharmacy8010034

**Published:** 2020-03-09

**Authors:** Piotr Merks, Damian Świeczkowski, Mikołaj Zerhau, Anna Gawronska, Anna Kowalczuk, Klaudiusz Gajewski, Ralf Däinghaus, Miłosz Jaguszewski, David Brindley

**Affiliations:** 1Faculty of Medicine, Collegium Medicum, Cardinal Stefan Wyszyński University, Warsaw 01-938 Poland; 2First Department of Cardiology, Medical University of Gdansk, Gdańsk 80-211, Poland; d.swieczkowski@o2.pl (D.Ś.); jamilosz@gmail.com (M.J.); 3dr. Józef Psarski Masovian Specialist Hospital in Ostrołęka, Ostrołęka 07-10, Poland; mikizerh@gmail.com; 4Standardisation Department, GS1 Expert in Healthcare, Poznan 61-755, Poland; Anna.Gawronska@ilim.poznan.pl; 5National Institute of Medicines, Warsaw 00-725, Poland; a.kowalczuk@nil.gov.pl; 6MedAspis GmbH, Düsseldorf, Germany; klaudiuszgajewski@gmail.com (K.G.); ralf.dainghaus@medaspis.com (R.D.); 7UCL Centre for the Advancement of Sustainable Medical Innovation, The University of Oxford, Oxford OX39DU, UK; david.brindley@paediatrics.ox.ac.uk

**Keywords:** decommissioning, falsified medicines directive, hospital pharmacy

## Abstract

**Background:** The introduction of a medicines verification and decommissioning system into the hospital pharmacy may result in an increased workload for pharmacy staff. The pilot implementation allows us to understand all the implications of the process, optimize process workflows, and estimate the time and cost of implementation. **Methods:** All the packages received at the hospital pharmacy had a 2D data matrix codes and were scanned. We analyzed the time needed to unpack a variety of products, scan them, and receive the notification. **Results:** In total, 144 packages were scanned at an average time of 3.05 s, with most (86.9%) under 4 s. Manual decommissioning using handheld scanners was less efficient than the automated solution tested and resulted in an additional 0.4 full-time equivalent hours per million packages per year. The pattern and total time of manual scanning depended not only on the quantity but also the size of the package and type of packing. **Conclusions:** This evaluation of scanning performance allows optimizing the process at operational, technical, and resource levels for medicine verification and decommissioning.

## 1. Introduction

The falsified medicines directive (FMD) introduced in 2011 by the European Union (EU) was aimed at harmonizing legislation across Europe, and introduced a common system preventing the entry of falsified medicinal products into the legal supply chain [1]. That was a response to an observed increase in the incidence of pharmaceutical counterfeiting [2,3,4]. Counterfeit medicines are difficult to identify in general. In Europe, combating the counterfeiting of pharmaceutical products evolved from a national level model into a shared responsibility model, involving manufacturers uploading unique identifiers into the European hub of the European Medicines Verification System and then decommissioning the medicines once delivered or dispensed, before being administered to the patient. From 2019, the authenticity verification process could take place at any point throughout the healthcare supply chain. In cases of the absence of the unique identifier, duplicate identifiers, or the use of the identifiers in another place or situation, the pharmacy has to be alerted and must take some specific action.

The introduction of a medicines verification and decommissioning system may increase the workload of secondary personnel in a pharmacy. It is essential to understand this workflow to be able to facilitate optimal implementation. Despite this, the 2019 deadline passed and the directive has been implemented in all pharmacies in the EU; so studying the process will help in developing best practices for medicine decommissioning and performing optimizations in the current setting. Here we present the results of a pilot study performed in a hospital pharmacy before the implementation of the directive, to evaluate target settings and the workload associated with decommissioning.

## 2. Materials and Methods

The study was performed in the hospital pharmacy of the Dr. Józef Psarski Masovian Specialist Hospital in Ostroleka, Poland. It is the main general hospital for a city of over 50,000 inhabitants and its surroundings. A diagnostic questionnaire was used to assess the type of pharmacy and the implications of the FMD. The hospital was considered large, with over 500 beds, which implied the need of the hospital pharmacy to dispense a high volume of products. The pharmacy was regularly supplied by wholesalers and manufacturers, and provided medicines to wards and collected them back. It would return medicines to the distributors for a variety of reasons. The pharmacy was not equipped with any automation or drug picking device. A point of authentication in the pharmacy was setup in the dispensing area. Upon receipt of a delivery, technicians checked the quantity of medicines received against the supplier’s delivery note. All the packages received by the hospital pharmacy with a 2D data matrix code were scanned with a handheld device. During the project, a real-life situation was simulated; as soon as the 2D data code was scanned, a response appeared. This data was analyzed to assess the frequency of scanning, the time needed to complete this activity and the response times. Just the amount of time needed for scanning was analyzed, without the time needed for quality control and identification of the medicines should they be counterfeit. This data was then used to assess if the verification and decommissioning of the medicinal products had an impact on the overall workload and the medicine dispensing process (within a secondary care environment) and to give a general overview of the outcomes of FMD application.

MedAspis B.V. (Amsterdam, The Netherlands) provided the technical solution necessary to complete the pilot. They provided a handheld battery-powered personal digital assistant (PDA) device with an inbuilt touchscreen and 1D/2D scanner and Internet connection via a 3G/4G network. The weight of the device, including the battery, was 240 g.

## 3. Results

During a single working day in June 2018, all the packages of medicines with a 2D data matrix code were scanned after delivery to the hospital pharmacy. In total, 144 packages of 27 different medicines were scanned. This included different quantities, package sizes, packaging types in orders.

The response time of the system varied from 0.01 to 0.20 s. With the handling time (e.g., opening of the cardboard box, removing any shrink film or rubber bands), the average (standard deviation) time needed to scan each package of medicines was 3.05 ± 02.02 s. The minimum time was below 1 s and the maximum was 16 s. Figure 1 presents the response times of scans of the same medications.

In most cases (86.9%), the time needed for this process was less than 4 s per package. Only a few scans required a longer time. Table 1 presents the estimation of workload in hours and full-time equivalents (FTEs) needed to verify or decommission the medicines. This was calculated based on the average time needed to verify/decommission one package that potentially falls under the delegated regulation and the number of packages dispensed annually. If the verification and decommissioning is done at goods-in and goods-out steps, and assuming that both processes needed a similar time, the numbers in Table 1 should be doubled.

Figure 2 shows the times between sequential scans of packages from single bundles of products with different quantities in delivery, size, and different mode of packing. Small numbers of packages can be scanned quickly in almost equal intervals (Figure 2a). A number of packages can be bound with rubber bands, and the removal time is visible during scanning as the single increase in time between scans (Figure 2b,c). Figure 2d,e represents the characteristics for products in cardboard boxes when a few packages were picked from the box for scanning. Finally, Figure 2f shows the scanning of a high number of large boxes of a product, each containing 100 ampules of a drug.

## 4. Discussion

Implementation of the FMD in the hospital pharmacy is a substantial challenge. Fulfillment of the FMD requirements implies additional step(s) and increased workload during delivery and accuracy checking (goods-in) before handing out medicines to wards (goods-out). Evaluation of the time spent in the dispensing areas of the pharmacy to verify and decommission large quantities of different types of medicines would help in optimizing the daily routine and the scale of the additional workload.

In the most simple setting of a single dispensing area where packages are scanned with a handheld device, the average time for scanning a package of a drug was 3.05 s. The decommissioning exercise at the point of arrival of delivery of packages into the hospital was the preferred option, with 75% of hospitals working according to this practice [5]. Verification and decommissioning of the unique EAN could be aided by the incorporation of a process using a robotic dispensing system [6]. In a setting of different types of dispensing points, including manual and automated points, the average time was measured to be around 2 s per pack [7]. Using logistic solutions such as aggregated codes (allowed by the FMD) [1] would ease the process, but requires the willingness of the manufacturers and distributors to provide these codes [5].

The verification activity requires a significant amount of time and adds extra costs, especially at those hospitals where medicine delivery checks were not part of the daily routine. For Poland, additional costs of introducing the FMD requirements in hospital pharmacies, evaluated by the European Alliance for Access to Safe Medicines, varied between 1.8 to 3.6 million euro. The minimal and maximal estimations were based on the time necessary for scanning a single pack ranging from 2.1 to 4.2 s [5]. The number of additional FTEs required by hospitals that may choose to verify the pack on first receipt (goods-in) and then decommission at the point of dispensing (goods-out) would face twice the increase in scanning time, which in the scenario studied here would equal an average of 6.1 s per pack and would double the amount of work and number of FTEs required as presented in Table 1. It should be noted that verification of the codes is not a mandatory operation for hospital pharmacies while decommissioning is.

Although in our study, random medicines were scanned based on a limited representation of packages with 2D data matrix codes at the time of the study, results have revealed that the amount of time for scanning a delivery did not depend just on the quantity of packs, but also on their size and type of packing. Commonly used methods at pharmacies, namely rubber bands, cardboard boxes, and large-size packages, have specific patterns of scanning times. Analysis of several patterns allowed us to recognize the typical manual manipulation steps required to accomplish scanning.

This study has some limitations. At the time of the study, the number of medicines available with the 2D data matrix codes was limited and only a small part of the delivered products were included. Just the time necessary to perform scanning was evaluated and did not include all the workload necessary to fulfill all FMD requirements. The study period was relatively short and to make a more representative statement, an extended study period is required. The presented findings and information can be applied to hospitals with similar infrastructure and logistics.

## 5. Conclusions

To understand the extent of additional FMD-related work, it is rational to perform an on-site evaluation of performance to optimize the current practice on operational, technical, and resource levels. Manual handling of medicines is less effective than automated/robotic settings.

## Figures and Tables

**Figure 1 pharmacy-08-00034-f001:**
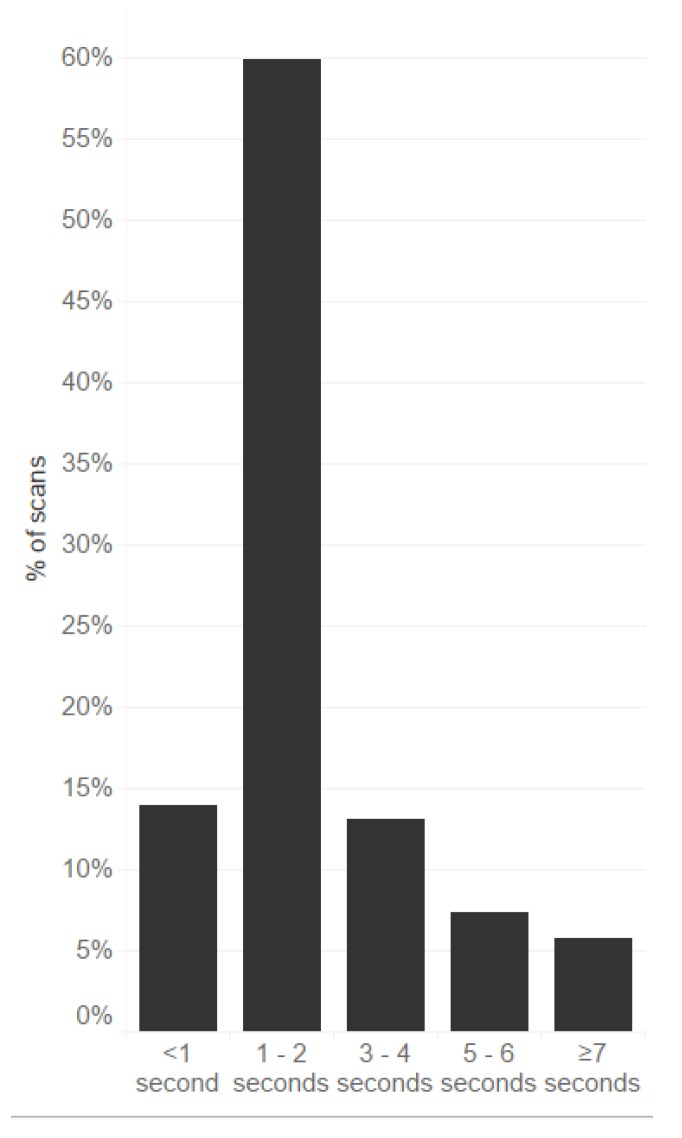
Percentage of intervals between subsequent scans of packages (*n* = 144) of medicines (27 different EANs).

**Figure 2 pharmacy-08-00034-f002:**
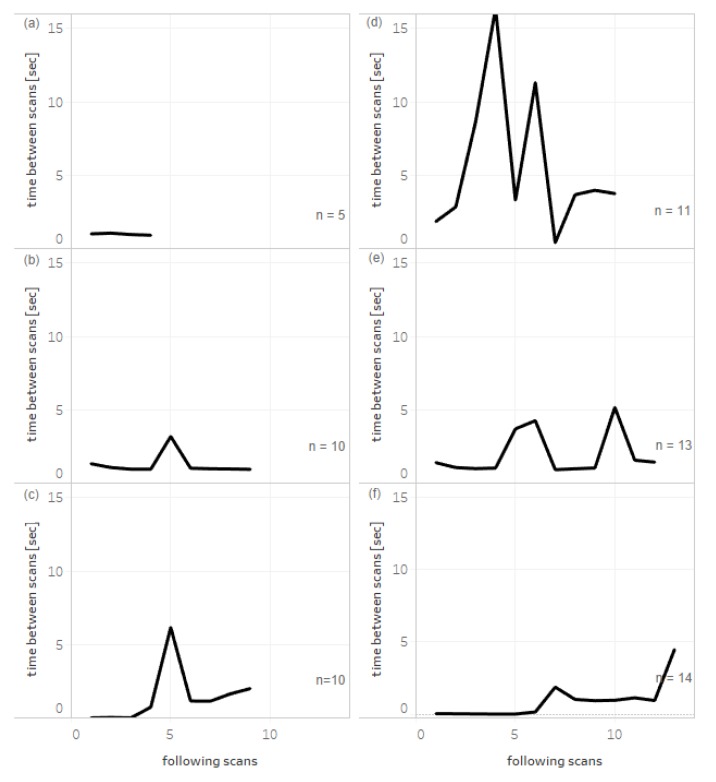
Example patterns of scanning of products with different quantities and types of packaging. Deliveries of products were scanned sequentially and the first data point indicates the interval between scanning the first and the second package. (**a**) Packages, each containing 30 tablets of bound product scanned without band removal. Packages, each containing 5 ampules (**b**) or 30 tablets (**c**) bound with a rubber band. Packages, each containing 10 ampules (**d**), and packages, each containing 60 tablets (**e**) packed into cardboard boxes. (**f**) Large packages of a drug (each containing 100 ampules). *n* indicates the number of single packs scanned of the same EAN code.

**Table 1 pharmacy-08-00034-t001:** Estimation of additional human resource to verify/decommission prescription packs in the hospital pharmacy depending on the number of packs dispensed annually.

Number of Packs Dispensed by Pharmacy Annually	Average Time of Verification or Decommission of a Pack [sec]	Additional Work Monthly [h]	Full-Time Equivalents ^1^ Required
1 million	3.05	71	0.4
0.75 million		53	0.3
0.5 million		35	0.2
0.25 million		18	0.1

^1^ Typically full-time equivalents of pharmacy technicians (full-time employee works 40 h/week).

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
