# Peer review of "Pilot Implementation of Falsified Medicines Directive in Hospital Pharmacy to Develop Best Practices for Medicine Decommissioning Process"

_pharmacy, 2020, doi:10.3390/pharmacy8010034_

Round 1

Reviewer 1 Report

Major comments.

In figure 1, the percentages are presented as averages. Can the authors include the standard deviations for the data. It is a bit unclear how the average verification calculated in Table 1 is the same regardless of the number of packs dispensed annually. Can the authors clarify how they came about this value?

Minor comments.

1. Line 43 has a typographic error, please correct sentence. 

Author Response

Dear Reviewer 1

Thank you for your comments. We calculated the standard deviation of scanning time and added it to the text. In the Table 1. the average time of scanning one pack (column 2) was used to calculate the number of hours monthly needed for scanning (column 3) and number of additional FTEs necessary to perform scanning (column 4), depending on the number of packages dispensed in a site (column 1).

Sincerly yours

Author

Reviewer 2 Report

Authors presented a pilot method for verification and decommissioning of marketed medicines before starting patient’s administration. The described set-up‘s aim was also to decrease the workload from hospitals and pharmacies. The implemented method seemed very useful as part of the staff’s workload was replaced by an automatized system.

However, authors presented just how the packages were checked-up but no mention on the quality control of the medicines was inserted; or the way authors described the methods did not refer also to the technical quality control.  More particularly, if a medicine is a subject of counterfeit, how the Authors can evidence this?

Introduction: The pilot method to scan the packages was it before implemented in another place in Poland or other country? Who invented the method or where the main idea started from? Authors should update the introduction part with more and newer literature. Additionally, at the end of introduction, Authors should insert a statement on their improvement within the field and what their device brings new.

Materials and methods: what parameters the pilot device can measure could be useful to be inserted. Authors are requested to provide details on the device performing the analysis.  

Results: Authors provided results on the scanning number and speed mostly; no details on other parameters measured; how the device is providing the final outcome; the mentioned 2 D pattern can have any representation in order to support the provided information?

Conclusion: conclusion should be updated as concerning the details asked above e.g. measured parameters.

Figure 2: Authors did not explain what “n” means or refer to; authors should insert supportive details.

Author Response

Dear Reviewer 2

Thank you for your comments. Our study aimed to evaluate the time necessary for scanning products in the real-world setting of hospital pharmacy. Since the routine of the pharmacy participated in the study was manual work, the scanning was implemented as part of that daily work. We did not aim to decrease the time but show how much it will take doing it manually. In the discussion, we confronted the result with other results coming from automated systems.

We described the device used in the study in more detail. However, we focused only on time between subsequent scans. The device reports information from 2D codes and result of verification and it is also stored in the database. Evaluation of this information was not its scope of our study and we indicated that it was a limitation in the discussion. Besides, we described in more detail how medicines' identity is verified at the European level.

We indicated that the n in Figure 2 indicates number of packages of the same EAN code scanned seusequently.

Sincerely yours,

Author

Reviewer 3 Report

This is a very interesting paper talking about an innovative approach; to outside readers - like myself - it is a  bit confusing to determine what you mean by decommissioning. I can understand it through some context clues but I was wondering if you could provide some specific definition.

Furthermore, to outside readers who may not be familiar with this process, perhaps a quick discussion about what the process is usually like and how hospitals go about doing this initiative if they do not have this type of technology - if this is something that is done. 

Were there also any positively identified counterfeit medications using this technology during the study process? If so, did the use of technology increase the amount of medications identified vs a manual process? What is the usual rate of counterfeit medications that appear in hospitals? 

I also noticed a few grammatical corrections - please see below

Introduction

  • Line 40 - need spacing between words "introducecommon"

Conclusion

  • Line 184 - need to add "is" in the last sentence
    • Manual handling of medicines is less effective...

Author Response

Dear Reviewer

Thank you for your comments. We improved the manuscript base on them. In ithe Introduction, we added explained the nature of decommissioning as the final step at the end of the supply chain when the product is supplied to the patient or leaves the pharmacy. Medicine can be verified as many times as it is needed, but decommissioning can be done only once, and upon it the unique identifier is deactivated. If scanned after decommissioning (e.g., in another place), it will inform that medicine can be falsified. In the study, we did not identify such medicines. We added information in results. Please note that in limitations of our study we indicated that time calculated is limited only to scanning and does not include other actions needed to fulfill requirements of the FMD (e.g., handling of medicines verified as falsified). We do not know what the frequency of such scans is, however, at the beginning of implementation, false-positive scans were quite frequent.

The manuscript was corrected fro language errors.

Kindest Regards,

Piotr Merks

Round 2

Reviewer 2 Report

The authors answered the requested queries. 

Author Response

Thank you for pointing attention. We performed proofreading of text and resubmit new version.

Kindest Regards,

Piotr Merks